# Land, buildings, and equipment acquisitions in U.S. hospitals: A fifteen-year perspective

**Kangkang Qi[1], Xuefeng Jiang[2], Ge Bai[3]\***

**1** Assistant Professor of Information Systems Management, Harbert College of Business, Auburn University, Auburn, AL United States of America, **2** Plante Moran Faculty Fellow, Professor of Accounting & Information Systems, Broad College of Business, Michigan State University, East Lansing, MI, United States of America, **3** Professor of Accounting, Johns Hopkins Carey Business School, Professor of Health Policy and Management, Johns Hopkins Bloomberg School of Public Health, Washington, DC, United States of America

\* gbai@jhu.edu

**Data Availability Statement:** The data underlying the results presented in the study are available from RAND (https://www.rand.org/pubs/tools/TL303.html).

**Funding:** Ge Bai received funding from Arnold Ventures. The funders had no role in study design,

## Abstract

Hospitals acquire and maintain long-term operating assets such as land, buildings, and equipment. In this study, we analyzed hospitals' long-term assets acquisitions data extracted from the Medicare Cost Report, a mandatory annual filing for all Medicare-certified hospitals. The first objective of this study is to examine the time trend of land, buildings, and equipment acquisitions of all general acute care hospitals in the U.S. from 2005 to 2019 to understand the relative magnitude and temporal changes for the operating assets. The second objective is to examine the 15-year accumulated acquisitions of land, buildings, and equipment per capita in each state to understand the variations of potential access to hospital operating resources across states. To understand the longitudinal changes in acquisitions of operating assets for each year from 2005 to 2019, we calculated the total acquisition amounts across all hospitals for land, buildings, and equipment, respectively, and adjusted the amounts to 2019 dollars based on the consumer price index (CPI). For each state (including Washington D.C.) and the whole nation, the 15-year accumulated CPI-adjusted acquisition amounts per capita for land, buildings, and equipment were also calculated, respectively. The nationwide acquisitions of those operating assets grew rapidly from 2005 to 2008 followed by a negative overall growth from 2008 to 2014 and since 2015, started increasing steadily again. In 2019, U.S. general acute care hospitals acquired $3.0 billion of land, $44.6 billion of buildings, and $33.9 billion of equipment. Huge geographical variation in per capita cumulative total asset investment were also found with the first place North Dakota having a per capita investment that is almost four times higher than that in the lowest ranked state of Alabama.

## Introduction

To deliver care to patients, hospitals acquire and maintain long-term operating assets such as land, buildings, and equipment. Underinvestment in these operating assets may limit patients' access to high-quality care. For example, the shortage of ventilators and inpatient beds limited many hospitals' ability to treat COVID-19 patients [1]. However, acquiring too many long-term assets may reflect inefficient capital investment decisions and wasteful health care

data collection and analysis, decision to publish, or preparation of the manuscript.

**Competing interests:** The authors have declared that no competing interests exist.

spending [2]. Despite the importance of hospitals' investments in long-term assets, no academic study, to our knowledge, has examined this issue, which is potentially attributable to the lack of accessible data that are comparable across hospitals.

In this study, we took advantage of the required reporting components in the Medicare Cost Report, a mandatory annual filing for all Medicare-certified hospitals, to obtain reliable and comparable information regarding hospitals' long-term assets acquisitions. This study has two main objectives. The first objective is to examine the time trend of land, buildings, and equipment acquisitions of all general acute care hospitals in the U.S. from 2005 to 2019 to understand the relative magnitude and temporal changes for the operating assets. The year 2005 is the second year after the Medicare Prescription Drug, Improvement, and Modernization Act was enacted, which substantially increased Medicare reimbursement to rural hospitals [3]. The year 2019 is the most recent year for which data is available.

The second objective is to examine the 15-year accumulated acquisitions of land, buildings, and equipment per capita in each state to understand the variations of potential access to hospital operating resources across states. Accumulated acquisitions reflect the sum of the market value (at the point of acquisition) of all assets obtained during a given period. We adjusted all acquisitions before 2019 for inflation to 2019 dollars. We conducted state-level analysis because laws and regulations, population changes, and demographics vary substantially across states, which may affect hospitals' operational and financial environment and may influence their long-term asset acquisitions decisions. A similar nationwide comparison of health care quality across states was conducted and reported by the Agency of Healthcare Research and Quality at the Department of Health and Human Services [4].

## Materials and methods

2005–2019 Medicare Cost Reports of all Medicare-certified general acute care hospitals in the United States were examined by using the RAND Hospital Data [5], a compiled and processed version of Medicare Cost Reports, published by the Centers for Medicare and Medicaid Service [6]. There were between 4,667 and 5,045 hospitals in each year and a total of 72,159 hospital-year observations in the dataset. Medicare Cost Reports contain the acquisition amounts for land, buildings, and equipment on Worksheet A-7 Part I. The acquisition amount includes the original acquisition and subsequent improvements (such as renovation, expansion, and upgrading).

To understand the longitudinal changes in acquisitions of operating assets for each year from 2005 to 2019, we calculated the total acquisition amounts across all hospitals for land, buildings, and equipment, respectively, and adjusted the amounts to 2019 dollars based on the consumer price index (CPI) [7]. We then plotted a 15-year trend for each type of asset and calculated the compound annual growth rates (CAGRs).

Next, for each state (including the District of Columbia) and the whole nation, the 15-year accumulated CPI-adjusted acquisition amounts per capita for land, buildings, and equipment were calculated, respectively. The accumulated acquisition per capita is the sum of all CPI-adjusted acquisitions amounts for a given asset type between 2005 and 2019, divided by the 2019 population in that state or the nation. Finally, we ranked all states and the District of Columbia based on their total accumulated operating assets per capita to understand the cross-state variation.

## Results

In 2019, U.S. general acute care hospitals acquired $3.0 billion of land, $44.6 billion of buildings, and $33.9 billion of equipment, representing 3.3%, 4.1%, and 2.4% CAGR, respectively, from 2005 (in 2019 dollars) (Fig 1: Land, Buildings, and Equipment Acquisitions by U.S.

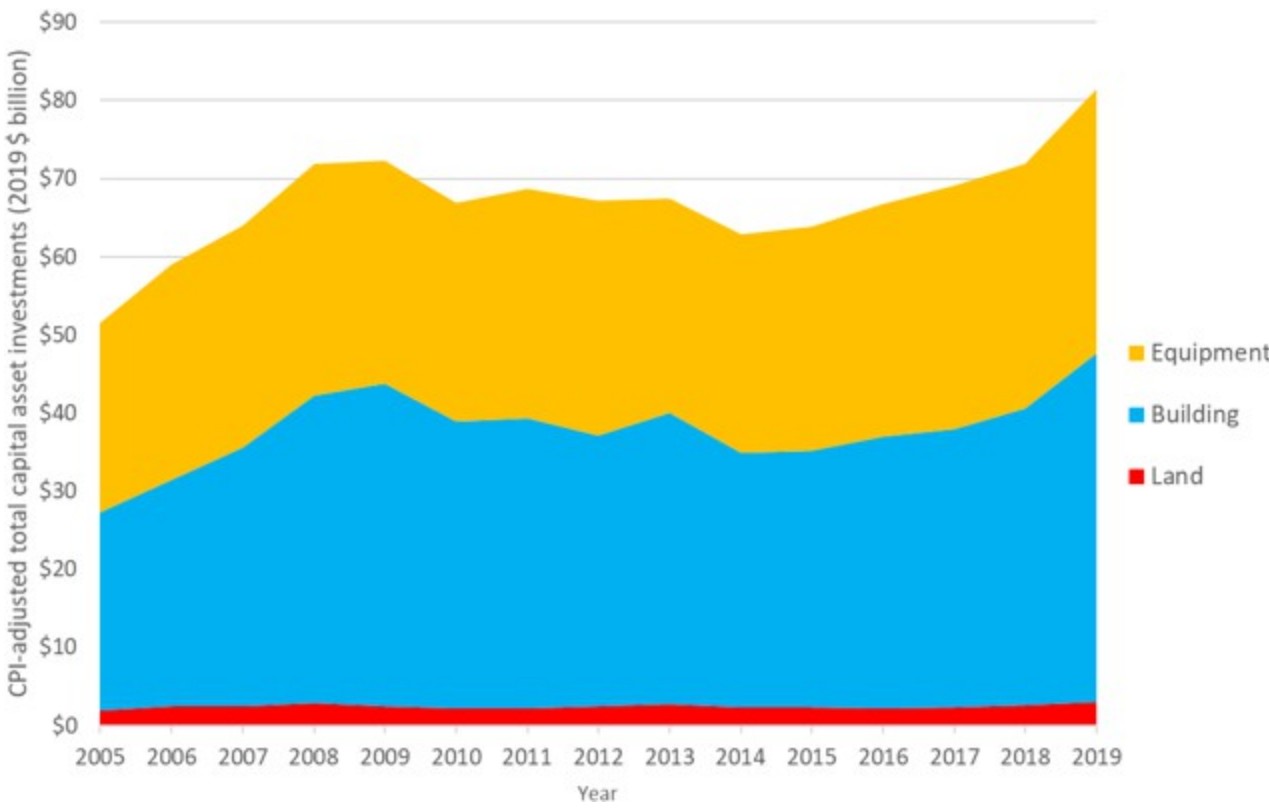

**Fig 1. Land, buildings, and equipment acquisitions by U.S. hospitals, 2005-2019[a].** Abbreviation: CPI: Consumer Price Index. [a] The sample includes all U.S. Medicare-certified general acute care hospitals in each year.

Hospitals, 2005–2019). The acquisitions grew rapidly from 2005 to 2008 (14.4% CAGR for land, 15.9% for buildings, and 6.9% for equipment). From 2008 to 2014, the acquisitions experienced negative overall growth (-3.5% CAGR for land, -3.1% for buildings, and -1.0% for equipment). Since 2015, the acquisitions have increased steadily (5.3% CAGR for land, 6.5% for buildings, and 3.9% for equipment). From 2005 to 2019, buildings acquisitions were greater than equipment acquisitions or land acquisitions, representing between 49% and 57% of total acquisitions of operating assets; equipment acquisitions fluctuated between 41% and 49%, and land acquisitions remained 3%-4% of the total purchases of operating assets.

The 2005–2019 accumulated acquisitions per capital of hospital operating assets were $3,063, including $112 of land, $1,620 of buildings, and $1,331 of equipment (in 2019 dollars) (Table 1: 2005–2019 CPI-adjusted Hospital Acquisitions of Land, Buildings, and Equipment per Capita, by State). Substantial variation existed across states. Colorado had the highest accumulated acquisition in land per capita ($279), while the District of Columbia had the lowest ($31). Massachusetts had the highest accumulated buildings acquisition per capita ($2,971), approximately four times as much as that in Nevada ($754), Alaska ($753), and Alabama ($718). North Dakota had almost $3,000 of accumulated equipment acquisition per capita (the highest state), while those of New Mexico, Arkansas, California, Hawaii, Nevada, and Alabama were below $1,000.

## Discussion

We conducted the first longitudinal analysis of all U.S Medicare-certified short-term general hospitals' acquisitions of land, buildings, and equipment and state-level variations in the

**Table 1. 2005–2019 CPI-adjusted Hospital Acquisitions of Land, Buildings, and Equipment per Capita, by State[a].**

| State (total $ ranking)[b] | Total | Land | Buildings | Equipment |
|---|---|---|---|---|
| North Dakota (1) | $5,537.2 | $162.9 | $2,397.6 | $2,976.7 |
| Massachusetts (2) | $4,662.9 | $74.2 | $2,970.7 | $1,618.1 |
| South Dakota (3) | $4,376.4 | $135.4 | $2,230.7 | $2,010.3 |
| West Virginia (4) | $4,353.6 | $126.3 | $1,793.8 | $2,433.5 |
| Nebraska (5) | $4,292.3 | $104.4 | $2,065.6 | $2,122.4 |
| Wisconsin (6) | $4,199.2 | $136.8 | $2,405.8 | $1,656.6 |
| Ohio (7) | $4,045.3 | $164.0 | $2,101.5 | $1,779.8 |
| Illinois (8) | $3,974.5 | $136.4 | $2,299.3 | $1,538.8 |
| Dist. of Columbia (9) | $3,967.3 | $30.9 | $1,789.5 | $2,146.9 |
| Montana (10) | $3,958.9 | $173.7 | $1,950.3 | $1,834.9 |
| Indiana (11) | $3,870.6 | $165.8 | $1,922.2 | $1,782.6 |
| Vermont (12) | $3,821.8 | $105.4 | $2,119.5 | $1,596.9 |
| Missouri (13) | $3,697.0 | $100.4 | $1,974.4 | $1,622.2 |
| Wyoming (14) | $3,675.3 | $189.4 | $1,708.7 | $1,777.2 |
| New York (15) | $3,658.0 | $74.6 | $2,005.2 | $1,578.2 |
| Delaware (16) | $3,579.5 | $86.6 | $1,552.8 | $1,940.1 |
| Maine (17) | $3,568.9 | $90.1 | $1,594.6 | $1,884.1 |
| Connecticut (18) | $3,542.5 | $100.0 | $1,959.5 | $1,482.9 |
| Iowa (19) | $3,527.0 | $133.4 | $1,779.4 | $1,614.2 |
| New Hampshire (20) | $3,476.3 | $78.3 | $1,659.0 | $1,739.0 |
| Colorado (21) | $3,475.1 | $279.0 | $1,886.4 | $1,309.7 |
| Idaho (22) | $3,464.9 | $143.2 | $1,976.5 | $1,345.1 |
| Pennsylvania (23) | $3,429.3 | $134.0 | $1,710.4 | $1,584.9 |
| Michigan (24) | $3,307.6 | $81.6 | $1,547.7 | $1,678.4 |
| Kentucky (25) | $3,273.5 | $99.9 | $1,739.4 | $1,434.1 |
| Kansas (26) | $3,247.5 | $106.8 | $1,686.3 | $1,454.4 |
| Maryland (27) | $3,089.8 | $111.5 | $1,702.6 | $1,275.7 |
| Minnesota (28) | $3,075.0 | $82.8 | $1,713.8 | $1,278.4 |
| Washington (29) | $2,997.3 | $128.7 | $1,593.5 | $1,275.1 |
| Virginia (30) | $2,933.3 | $94.2 | $1,517.0 | $1,322.2 |
| New Jersey (31) | $2,860.5 | $86.9 | $1,557.8 | $1,215.8 |
| Louisiana (32) | $2,799.7 | $124.9 | $1,386.7 | $1,288.2 |
| Oregon (33) | $2,792.8 | $85.0 | $1,703.8 | $1,004.0 |
| California (34) | $2,753.0 | $142.6 | $1,686.9 | $923.5 |
| Georgia (35) | $2,702.3 | $120.7 | $1,211.9 | $1,369.7 |
| North Carolina (36) | $2,676.3 | $84.2 | $1,446.2 | $1,145.8 |
| Florida (37) | $2,665.2 | $118.0 | $1,248.0 | $1,299.3 |
| Texas (38) | $2,548.9 | $82.7 | $1,382.2 | $1,084.1 |
| Mississippi (39) | $2,540.2 | $100.4 | $1,197.1 | $1,242.7 |
| Utah (40) | $2,516.4 | $118.6 | $1,154.7 | $1,243.1 |
| Tennessee (41) | $2,457.0 | $86.8 | $1,109.2 | $1,261.0 |
| South Carolina (42) | $2,398.0 | $119.3 | $1,083.4 | $1,195.2 |
| Oklahoma (43) | $2,387.6 | $69.6 | $1,116.6 | $1,201.4 |
| Arizona (44) | $2,238.3 | $88.3 | $966.6 | $1,183.4 |
| Rhode Island (45) | $2,198.2 | $58.3 | $820.9 | $1,319.1 |
| New Mexico (46) | $2,109.4 | $77.5 | $1,048.5 | $983.4 |
| Arkansas (47) | $2,025.9 | $79.2 | $1,013.5 | $933.3 |

*(Continued)*

**Table 1.** (Continued)

| State (total $ ranking)[b] | Total | Land | Buildings | Equipment |
|---|---|---|---|---|
| Hawaii (48) | $1,972.3 | $67.6 | $988.8 | $915.9 |
| Alaska (49) | $1,920.1 | $70.2 | $752.8 | $1,097.1 |
| Nevada (50) | $1,658.2 | $56.9 | $753.9 | $847.4 |
| Alabama (51) | $1,561.5 | $48.4 | $717.7 | $795.4 |
| **Nationwide**[c] | $3,062.8 | $111.6 | $1,620.5 | $1,330.7 |

Abbreviation: CPI: Consumer Price Index.

[a] The sample includes all U.S. Medicare-certified general acute care hospitals in each year from 2005 to 2019.

[b] Acquisition amount per capita for each state was measured as the summation of the CPI-adjusted acquisition amounts from 2005 to 2019 in the state divided by the 2019 state population for land, buildings, and equipment collectively or respectively.

[c] Acquisition amount per capita nationwide was measured as the summation of the CPI-adjusted acquisition amounts from 2005 to 2019 in the U.S. divided by the 2019 U.S. population for land, buildings, and equipment collectively or respectively.

15-year accumulated acquisitions of these assets per capita. We found that between 2005 and 2019, the amounts of buildings and equipment acquisitions were comparable (the former was greater every year), while land acquisitions accounted for only 3–4% of the total acquisitions of operating assets. There are at least two potential explanations. First, in most geographic areas, land value is relatively small compared to the value of buildings and equipment needed to operate a modern hospital. Second, hospitals only need to purchase land once but have to invest in maintenance and improvement on buildings and equipment annually. In other words, once a hospital has been built, except for occasional facility expansions that need a land purchase, continuous improvements on a hospital's functionality rarely involve land but routinely involve buildings and equipment.

After adjusting for inflation, we found that the acquisitions made by general acute hospitals in the U.S. increased sharply between 2005 and 2008 but dropped between 2008 and 2014. This pattern might be at least partially attributable to the 2008 financial crisis, which significantly affected hospitals' financial health and may have led to their reduced interest in acquiring operating assets. We also found a steady growth in operating asset acquisitions since 2015, a trend that might be associated with hospitals' improved financial positions and potentially Medicaid expansions implemented in many states since 2014.

Substantial variations existed in accumulated acquisitions of operating assets per capita across states. There are at least three potential explanations. First, acquisition values of land and buildings are influenced by local market characteristics, such as between rural and metropolitan locations. Second, states experienced different population growth during 2005–2019 and may have been subject to different needs for hospital asset acquisitions. Third, variations in state-level public health policies may have contributed to varying levels of hospital revenue and financial status. For example, Medicaid expansion decisions and the level of federal financial support differ substantially across states and can affect hospitals' various operational and investment decisions.

This study has several limitations. First, the detailed types of operating assets, such as the characteristics of equipment acquired, are unavailable in the data, which prevented us from conducting more specific asset acquisition analysis or understanding depreciation-related issues. Although our study examines the annual acquisitions and their longitudinal changes, our focus is narrowly confined to these asset-related cash outflows and the results cannot be used to accurately determine the current state of asset stock. This is because the annual disposal and depreciation amounts were not analyzed and, more importantly, the market value

information of asset stock is not available. For example, low, zero, or negative growth of acquisition does not necessarily indicate a shrinking asset stock, since the disposal and depreciation might have experienced negative growth and the market value of some assets may have appreciated.

Second, the acquisition value of land and buildings is influenced by geographic factors and thus has limited comparability across geographic regions. Third, we conducted the analysis of accumulated operating assets at the state level to take into consideration of policy variations across states. Future research can extend this research to explore potential causal relationships between health policy and operating asset acquisitions or to analyze other geographic factors to examine more granular variations in operating asset acquisitions. Fourth, the 2019 populations we used to derive the accumulated acquisitions of operating assets per capita do not take into account of the population changes in each state over time. Finally, this study, limited by the lack of data availability and its descriptive nature, cannot assess the optimality of operating asset acquisitions or provide evidence to support any causal relationships, which are promising areas for future research.

## Conclusion

This descriptive study documented land, buildings, and equipment acquisitions by U.S. general acute care hospitals from 2005 to 2019 and analyzed the 15-year accumulated land, buildings, and equipment acquisitions per capita in each state (based on 2019 population). Research on hospital quality of care and access to hospital operating resources may consider operating asset acquisitions as an influencing factor. In addition, policymakers interested in health equity should be aware of the geographic variations of operating asset acquisitions.

## Acknowledgments

All authors declare no potential conflicts of interest with respect to the research, authorship, or publication of this article. Ge Bai received funding from Arnold Ventures. The funders had no role in study design, data collection and analysis, decision to publish, or preparation of the manuscript. Ge Bai is also a visiting scholar at the Congressional Budget Office (CBO). This paper has not been subject to CBO's regular review and editing process. The views expressed here should not be interpreted as CBO's.

## Author Contributions

**Conceptualization:** Xuefeng Jiang, Ge Bai.

**Data curation:** Kangkang Qi.

**Formal analysis:** Kangkang Qi, Ge Bai.

**Methodology:** Kangkang Qi.

**Resources:** Ge Bai.

**Software:** Kangkang Qi.

**Supervision:** Xuefeng Jiang, Ge Bai.

**Writing – original draft:** Ge Bai.

**Writing – review & editing:** Xuefeng Jiang, Ge Bai.

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
