## [Decision Letter · Decision Letter 0]

15 Jun 2022

PONE-D-21-27752Land, Buildings, and Equipment Acquisitions in U.S. Hospitals: A Fifteen-Year PerspectivePLOS ONE

Dear Dr. Bai,

Thank you for submitting your manuscript to PLOS ONE. After careful consideration, we feel that it has merit but does not fully meet PLOS ONE’s publication criteria as it currently stands. Therefore, we invite you to submit a revised version of the manuscript that addresses the points raised during the review process.

The reviewer has provided some minor comments that should be addressed. These include the need to further discuss some of the limitations of the study. Their comments can be viewed in full, below.

We look forward to receiving your revised manuscript.

Kind regards,

Natasha McDonald, PhD

Associate Editor

PLOS ONE

Journal Requirements:

Reviewers' comments:

Reviewer's Responses to Questions

**Comments to the Author**

1. Is the manuscript technically sound, and do the data support the conclusions?

Reviewer #1: Yes

2. Has the statistical analysis been performed appropriately and rigorously? 

Reviewer #1: Yes

3. Have the authors made all data underlying the findings in their manuscript fully available?

Reviewer #1: Yes

4. Is the manuscript presented in an intelligible fashion and written in standard English?

Reviewer #1: Yes

5. Review Comments to the Author

Reviewer #1: This is a very straightforward and informative paper on asset acquisition by US hospitals between 2005 and 2019. The authors present basic data on overall trends and aggregate acquisition per capita by state. The analyses are pretty basic but appropriate for questions asked and for the nature of the data available to the authors. The results are purely descriptive, yet novel considering there’s been very little research in this specific area. The limitations are many, however they are adequately acknowledged. Overall, I think the contribution of the new data is significant and will be of interest to many readers. My only suggestion is that the authors discuss asset depreciation. It appears the data may not be available but the bounds it imposes on the interpretability of the results should be discussed. With the net growth in acquisition in single digits in the recent years (even negative in some cases), what can we say about the current asset stock especially for high-depreciation assets like equipment. At least a mention in the limitations section is warranted.

6. PLOS authors have the option to publish the peer review history of their article (what does this mean?). If published, this will include your full peer review and any attached files.

Reviewer #1: No

---

## [Author Response · Author response to Decision Letter 0]

26 Jun 2022

Please refer to the attached response letter for more details.

---

## [Decision Letter · Decision Letter 1]

6 Jul 2022

PONE-D-21-27752R1Land, Buildings, and Equipment Acquisitions in U.S. Hospitals: A Fifteen-Year PerspectivePLOS ONE

Dear Dr. Bai,

Thank you for submitting your manuscript to PLOS ONE. After careful consideration, we feel that it has merit but does not fully meet PLOS ONE’s publication criteria as it currently stands. Therefore, we invite you to submit a revised version of the manuscript that addresses the points raised during the review process.

 Please note that the previous reviewer has assessed your revised manuscript and suggested a few changes. Please address these when revising your submission. 

We look forward to receiving your revised manuscript.

Kind regards,

Carla Pegoraro

Division Editor

PLOS ONE

Journal Requirements:

Reviewers' comments:

Reviewer's Responses to Questions

**Comments to the Author**

1. If the authors have adequately addressed your comments raised in a previous round of review and you feel that this manuscript is now acceptable for publication, you may indicate that here to bypass the “Comments to the Author” section, enter your conflict of interest statement in the “Confidential to Editor” section, and submit your "Accept" recommendation.

Reviewer #1: All comments have been addressed

2. Is the manuscript technically sound, and do the data support the conclusions?

Reviewer #1: Yes

3. Has the statistical analysis been performed appropriately and rigorously? 

Reviewer #1: Yes

4. Have the authors made all data underlying the findings in their manuscript fully available?

Reviewer #1: No

5. Is the manuscript presented in an intelligible fashion and written in standard English?

Reviewer #1: Yes

6. Review Comments to the Author

Reviewer #1: Thanks for adding the text about stock vs flow in the limitations. It appears, through, that the last sentence of the new text seemingly contradicts the earlier part, in saying that zero acquisition necessarily means that stock is declining while just before it talks about market value appreciation. The last sentence could be clarified further or removed.

7. PLOS authors have the option to publish the peer review history of their article (what does this mean?). If published, this will include your full peer review and any attached files.

Reviewer #1: **Yes: **Olga Yakusheva

---

## [Author Response · Author response to Decision Letter 1]

16 Jul 2022

Please refer to the submitted response letter for details. Thank you.

---

## [Editor Report · Decision Letter 2]

19 Jul 2022

Land, Buildings, and Equipment Acquisitions in U.S. Hospitals: A Fifteen-Year Perspective

PONE-D-21-27752R2

Dear Dr. Bai,

We’re pleased to inform you that your manuscript has been judged scientifically suitable for publication and will be formally accepted for publication once it meets all outstanding technical requirements.

Kind regards,

Carla Pegoraro

Division Editor

PLOS ONE
---

## [Editor Report · Acceptance letter]

26 Jul 2022

PONE-D-21-27752R2 

Land, Buildings, and Equipment Acquisitions in U.S. Hospitals: A Fifteen-Year Perspective 

Dear Dr. Bai:

I'm pleased to inform you that your manuscript has been deemed suitable for publication in PLOS ONE. Congratulations! Your manuscript is now with our production department. 

Kind regards, 

on behalf of

Dr Carla Pegoraro 

Staff Editor

PLOS ONE